# Maintenance of Alveolar Ridge Dimensions Utilizing an Extracted Tooth Dentin Particulate Autograft and Platelet-Rich fibrin: A Retrospective Radiographic Cone-Beam Computed Tomography Study

**DOI:** 10.3390/ma13051083

**Published:** 2020-02-29

**Authors:** Snjezana Pohl, Itzhak Binderman, Jelena Tomac

**Affiliations:** 1Department of Oral medicine and Periodontology at University of Rijeka, Private Clinic Rident, 51000 Rijeka, Croatia; 2Department of Oral Biology, School of Dental Medicine and Department of Biomedical Engineering, Tel Aviv University, 6997801 Tel Aviv, Israel; Binderma@post.tau.ac.il; 3Department of Histology and Embryology at University of Rijeka, 51000 Rijeka, Croatia; jelena.tomac@medri.uniri.hr

**Keywords:** dentin graft, CBCT imaging, alveolar ridge preservation, bone substitutes

## Abstract

This study utilized radiographic comparative analysis in order to evaluate dimensional ridge changes four months after tooth extraction and immediate grafting with mineralized dentin particulate autograft and chopped platelet-rich fibrin. Fifty-eight extraction sockets with up to 2 mm of missing buccal bone in the coronal aspect compared to the lingual bone were included. Graft material was covered with either a platelet-rich fibrin membrane or collagen sponge with no effort to achieve primary closure. The dimensional changes of the ridge were assessed on cone-beam computed tomography (CBCT) images acquired prior to extraction and four months later. The reduction in the buccal bone plate thickness 1 mm, 3 mm, and 5 mm below the buccal crest was −0.87 ± 0.84 mm, −0.60 ± 0.70 mm, and −0.41 ± 0.55 mm, respectively. The mean ridge width changes 1 mm, 3 mm, and 5 mm below the crest were −1.38 ± 1.24 mm, −0.82 ± 1.13 mm, and −0.43 ± 0.89 mm, respectively. The average mid-buccal bone height gain was +1.1%, while the mid-lingual height gain was 5.6%. A mineralized dentin autograft with platelet-rich fibrin is effective in preserving post-extraction alveolar ridge dimensions.

## 1. Introduction

Dimensional alveolar bone volume decreases may occur up to 12 months after tooth removal with two-thirds of the respective bone volume loss occurring within the first three months of the healing period [1]. Findings from a systematic review indicate an overall ridge width decrease of 29.0–63.0% and a decrease in ridge height between 11.0% and 22.0% within the first six months after tooth extraction [2]. Likewise, the results of another systematic review indicated that bone remodeling during the post-extraction period demonstrates a mean horizontal bone loss of 3.87 mm in relation to a mean height loss of 1.67 mm [3]. Most of the buccal alveolar bone is bundle bone; therefore, it resorbs once losing its function after extraction [4]. In a prospective study based on two consecutive cone-beam computed tomography (CBCT) scans, it was shown that severe buccal bone resorption is correlated with a buccal bone wall thickness of less than 1 mm at socket type 1 [5].

Current findings regarding the success of different grafting procedures of extraction sites for ridge preservation show a lesser alveolar ridge dimensional loss compared to non-grafted sites [6]. Recent studies and clinical experience show that a partial extraction and retention of the buccal root fragment, attached to buccal bundle bone by a healthy periodontal ligament (PDL) and by fibers and cells to marginal gingiva, is a treatment modality capable of almost completely maintaining alveolar ridge dimensions [7,8,9]. Nonetheless, partial extraction therapy is contraindicated for teeth with mobility and periodontal-compromised teeth with pockets. However, mobile teeth, periodontal-diseased teeth, and teeth with an acute inflammatory process that are indicated for extraction can still be utilized as graft material for the post-extraction socket. It is beneficial to utilize extracted teeth as autogenous grafting material, especially as their processing is now efficient and simple, rather than discarding them as waste [8]. In their particulate form (graft), they are identified in the same way as an autogenic cortical membranous bone matrix since they were formed from the same neural crest cells. Moreover, dentin particles undergo ankylosis by bone deposited directly on the cementum and dentin, just like in ankylosed re-implanted avulsed teeth [10,11]. A human periodontal ligament fibroblast cell line showed a very promising growth reaction to the mineralized dentin in a recently published in vitro study, confirming its excellent biocompatibility [12]. The mineral and organic matrix compositions of a tooth, dentin, and cementum are almost identical to membranous bone [13], although, compared to bone, teeth contain more mineral. This feature may be beneficial for volume maintenance since at the remodeling stage dentin graft is resorbed slowly. As a supplement of bone graft material, the use of platelet concentrates such as platelet-rich plasma (PRP), platelet-rich fibrin (PRF), and concentrated growth factors (CGF), which regulate inflammation and angiogenesis, could be beneficial [14]. These biological additives were identified as producers of growth factors that promote the wound healing and regenerative process [15,16].

In our study, a second-generation platelet concentrate, i.e., leucocyte and platelet-rich fibrin (PRF), as introduced by Choukroun and co-workers was utilized as supplementary to the particulate dentin graft [17].

Recently, a human pilot study compared dimensional ridge changes eight and 16 weeks after tooth extraction and either filling the socket with a mineralized particulate dentin autograft (MPDA) (test group) or without grafting (control group), showing significantly better results for the test group [18].

The aim of this study was to evaluate the MPDA/PRF scaffold properties on a larger number of post-extraction sites than that of previous studies.

## 2. Materials and Methods

### 2.1. Study Design

In this single-arm, retrospective study, measurements were acquired via CBCT scans before tooth extractions and socket grafting and four months (±3 weeks) afterward. This analysis was observed in full accordance with the World Medical Association Declaration of Helsinki. The patients involved in this study were informed that data obtained during routine visits would be collected and used in an anonymous manner for statistical analysis; they signed written consent to participate. The patients’ specific files and data were kept confidential. The extracted data were assigned random case numbers. The Ethical Committee of the Clinic Rident approved the data collection. This retrospective study is reported according to the STROBE statement.

### 2.2. Participants and Treatment Protocol

Participants were consecutive individuals, requiring extraction of at least one hopeless non-root canal-treated tooth, who were candidates for treatment with dental implants and patients at the Clinic Rident in Rijeka, Croatia. Patients, treated by one surgeon (S.P.) between May 2017 and January 2019, were healthy adults (over 18 years of age) with no contraindications for implant placement, who returned in four months for the follow-up CBCT and implant placement.

Post-extraction sockets with no more than 2 mm of missing buccal bone in the coronal aspect compared to the lingual bone were included. Sockets with periradicular defects were not excluded. Sockets missing a buccal bone wall were not included in this study but will be analyzed separately instead.

Indications for tooth extraction were periodontitis, advanced caries, root fracture, and prosthetic reasons. In total, 13 patients with 61 dentin-grafted post-extraction sockets were included.

### 2.3. Surgical Procedure

Under local anesthesia, teeth were extracted with great care to preserve the buccal bone plate and the surrounding soft and hard tissues. The flap was not elevated. Care was taken to perform a thorough socket debridement.

A dentin autograft was prepared according to manufacturer recommendations [8]. Extracted teeth were thoroughly, mechanically cleaned utilizing high-speed carbide bur. All filling materials, calculus, decay, PDL, discolored dentin, and part of the enamel were removed before grinding. Clean teeth were dried by an air syringe and ground in a sterile chamber of the Smart Dentin Grinder® (SDG) unit (KometaBio Inc., Cresskill, NJ, USA). The SDG unit was programmed to collect 300–1200 μm particles in the collection tray.

The particulate teeth were immersed in a basic alcohol cleanser in a sterile container for 5 min to dissolve all organic remnants and bacteria, then dehydrated with sterile gauze. The particles were then rinsed twice with sterile phosphate-buffered saline solution. The platelet-rich fibrin (PRF) membranes were prepared according to the slow centrifugation protocol [19]. PRF membranes were pressed in a PRF box after blood centrifugation at 1300 rpm for 8 min in Choukroun’s PRF DUO centrifuge (Process for PRF, Nice, France). Sockets were grafted with a mixture of particulate dentin autograft (2/3) and chopped PRF membranes (1/3). Graft material was placed into the socket and covered either with PRF membranes or a collagen sponge (BEGO Collagen Fleece, BEGO, Bremen, Germany) to contain the particulate material. Absorbable monofilament sutures 5.0 (Surgicryl Monofast, SMI, St. Vith, Belgium) were utilized to affix the PRF membrane/collagen sponge over the grafted socket (Figure 1). No effort was made to achieve primary closure of the socket opening.

### 2.4. Radiographic Examination

CBCT scans were taken prior to tooth extraction using three-dimensional (3D) Promax (Planmeca OY, Helsinki, Finland). A CBCT scan was taken with a resolution of 0.3 mm (scan time: 8.5 s, exposure time: 4 s, 120 kV, 5 mA). Four months post extraction, all patients were recalled for a follow-up CBCT scan using the same settings as described above in preparation for the next step in the restoration treatment plan. All radiographic measurements and analyses were performed by one external examiner (WI, 3DVision, Cairo, Egypt).

#### 2.4.1. Buccal Bone Plate Thickness

Buccal bone plate thickness (BBT) was measured on baseline CBCT scans at three points (1 mm, 3 mm, and 5 mm) below the highest point of the buccal ridge perpendicular to the vertical reference line. The vertical reference line was outlined from the apex through the center of the socket (Figure 2a).

#### 2.4.2. Ridge Width and Height Changes

Ridge height and width measurements were acquired on the baseline and follow-up CBCT scans (Figure 3b–f). Horizontal and vertical reference lines were determined on the baseline CBCT image and duplicated onto the follow-up CBCT image as previously described [20]. In short, the vertical reference line was outlined from the apex through the center of the socket and the horizontal reference line through the apex perpendicular to the vertical line (Figure 2a).

The ridge width was measured parallel to the horizontal reference line 1 mm, 3 mm, and 5 mm from the highest buccal ridge point on baseline (RW-1, RW-2, RW-3, respectively) and follow-up (RW`-1, RW`-3, and RW`-5, respectively) CBCT images as shown in the drawing (Figure 2a,b). Dimensional ridge width changes were assessed based on the measurements performed at the baseline and after four months at three levels below the crest (ΔRW-1, ΔRW-3, and ΔRW-5, respectively).

Baseline mid-buccal and mid-lingual ridge heights (BH1 and LH1, respectively) (Figure 2a) and those during follow-up (BH2 and LH2, respectively) (Figure 2b) were measured from the most apical point of the alveolar socket parallel to the vertical reference line. The most coronal aspect on the buccal and lingual sides of the ridge at follow-up was determined as the coronal reference point regardless of whether it was a cortical bone plate or grafting material still in the osseointegration phase. Dimensional ridge height changes were assessed based on the measurements performed at the two samples on the buccal (ΔBH) and lingual (ΔLH) aspects.

#### 2.4.3. Buccal Bone Plate Reduction

The buccal bone plate reduction (BPR) was measured directly on the superimposed follow-up and baseline CBCT scans. To measure the BPR, the follow-up CBCT image was superimposed on the baseline CBCT image and precisely aligned by anatomical landmarks utilizing OnDemand3D App software (Seoul, Korea). Point-to-point distances between the two buccal surfaces with the respective angle to the vertical reference line were measured 1 mm, 3 mm, and 5 mm from the highest point of the buccal ridge determined on the follow-up CBCT image (BPR1, BPR3, and BPR5, respectively) (Figure 2c). Furthermore, dimensional changes at the molar sites were assessed for the mesial root in the lower jaw and the mesio-buccal root in the upper jaw.

### 2.5. Histologic Examination

Histological samples were harvested with a trephine bur during implant preparation, preserved in 10% neutral buffered formalin, and sent for histological and immunohistochemical analysis to the Department of Histology and Embryology Faculty of Medicine, University of Rijeka. Surgically obtained parts of alveolar bone were fixed immediately in 4% paraformaldehyde for 24 h and decalcified in Osteodec (Bio Optica 05-MO3005) for six days at room temperature.

Following several rinses in PBS, specimens were dehydrated through an alcohol sequence, cleared in xylene, and embedded in paraffin. Microtome sections (3 μm) were intended for histological and immunohistochemistry analysis.

The morphology of the dental and bone tissues was qualitatively evaluated on the hematoxylin and eosin-stained slides (Termo Fischer Scientific, Runcorn, UK).

Previously to the immunohistochemistry staining, endogenous peroxidase activity was inhibited by Dako Peroxidase-Blocking solution for 5 min at room temperature. Subsequently, sections were subjected to antigen retrieval with citrate phosphate buffer (pH 6.0).

To evaluate osteoblast differentiation and bone formation, primary antibodies against Osterix (Anti-Sp7/Osterix antibody-Chip Grade, ab22552, Abcam) were used, followed by polyclonal goat anti-rabbit/biotinylated secondary antibody (E 0432; Dako, Glostrup, Denmark) and Streptavidin-peroxidase (POD) conjugate (Roche Diagnostic GmbH; Mannheim, Germany). DAB (3,3′-Diaminobenzidine) was used as a chromogen (Dako) and counterstaining was performed with Shandon hematoxylin.

Slides were analyzed on an Olympus BX51 microscope and images were acquired by Olympus digital camera DP71 (Tokyo, Japan).

### 2.6. Statistical Analysis

Statistical analyses were performed using Social Science Statistics Online verified against SPSS Software version 25, IBM Corporation. Means and standard deviations of each of the measurement array datasets were calculated. One-way analysis of variance was applied for the comparison of the means, assuming a level of significance of 95% (*p* < 0.05). A hypothetical mean basis was determined based on two of the reference studies highlighted in Section 1. The hypothetical mean basis was determined as the average of the figures these studies provided. A highly significant *t*-test indicates that the difference between the sample and the expected value is significantly different; therefore, the null hypothesis shall be rejected.

## 3. Results

CBCT images from 12 patients who underwent a total of 58 extractions, where autologous particulate dentin processed from the extracted tooth and mixed with chopped platelet-rich fibrin was grafted at the same extraction site, were evaluated. The average age was 51 ± 14 years; nine out of 12 patients were smokers. In total, 13 patients with 61 MDPA/PRF grafted post-extraction sockets were included. One patient’s post-operative CBCT scan with three dentin-grafted sockets did not fulfill the CBCT quality level required for the type of measurements employed because of the artefacts caused by metals in adjacent areas and was, therefore, excluded.

In the mandible, 19 teeth were extracted, and the sockets were grafted, whereas 39 teeth were extracted in the maxilla. Among the extracted teeth, there were 22 incisors, 12 canines, nine first premolars, 10 second premolars, and five molars. Four incisors, two premolars, and one molar tooth presented periradicular defects. No adverse events were recorded during the four months after grafting.

Measurements of the initial buccal bone plate thickness and the horizontal buccal bone reduction four months after tooth extraction and grafting are shown in Table 1. Ridge width changes are shown in Table 2, and measurements of the buccal and lingual bone height from the tooth apex reference line up to the crest are shown in Table 3. In Table 4 a summary of statistical analysis findings for ridge width and height are compared to literature reference data highlighted in introduction.

Figure 4 shows typical ridge dimension changes for the periodontal compromised site (also see Figure 1). There is a gain in mid-palatal bone height (measured to the top of the graft still in the consolidation phase) and a loss of ridge width. Figure 2c graphically illustrates average dimensional changes. The green color denotes a fictive initial ridge composed of the mean buccal and lingual bone height, mean BBT, and ridge width, as displayed in Table 1, Table 2 and Table 3. The blue color represents the contours of the average fictive ridge, based on the measurements of the parameters mentioned above, 4 months after tooth extraction and socket grafting.

Histological analysis revealed new bone formation in close contact to dentin particles (ankylosis) with no signs of inflammation or fibrous encapsulation of the autologous augmentation material (Figure 5). New bone was confirmed by osteoblasts marked by antibodies against Osterix (Anti-Sp7/Osterix antibody-Chip Grade, ab22552, Abcam, Cambridge, UK) (Figure 6).

## 4. Discussion

Alveolar ridge preservation is a common clinical procedure performed to limit alveolar ridge resorption after tooth extraction to enable later implant placement or to preserve the pontic site. An ideal grafting material is biocompatible, osteoconductive/osteoinductive, and biodegradable with mechanical properties that enable the preservation of the alveolar contour, porosity, and surface properties that result in angiogenesis and finally bone formation. Furthermore, the material should be easy for chair-side use and cost-effective [21].

When autogenic dentin is grafted, the new bone deposited directly on the grafted surfaces will create a mineralized matrix connectivity between the host bone, new bone, and the grafted material (ankylosis) during the regenerative phase as shown in presented histology and in the literature [22,23]. In a recent study, a comparison of samples at different time points (four, five, and six months after MPDA/PRF grafting) showed a progressive increase in the proportion of bone with a decrease in the proportion of dentin [24]. Mazor et al. observed the same dynamic, with a 63% relative percentage of bone after seven months and a negligible residual mineralized dentin particle content [25].

From a clinical point of view, the most important ridge dimension changes after tooth extraction are buccal bone height changes, buccal bone resorption, and ridge width changes 1 mm below the crest. Maintaining buccal bone height is essential, especially for esthetic areas. Diminished vertical buccal bone dimension may result in recession. A human pilot study showed a vertical height loss of −4.2% for six MPDA grafted sockets compared to −16.7% for non-grafted sockets [18]. A recently published study showed less vertical BPR for sites grafted with dentin compared to a xenograft: −1.14 mm ± 0.81 mm buccal bone height reduction for the sockets grafted with deproteinized bovine bone with collagen, −0.97 mm ± 0.37 mm reduction for a demineralized dentin graft, and −0.82 mm ± 0.36 mm reduction for sockets grafted with demineralized dentin with bone morphogenetic protein 2 [26]. In an animal experiment, it was shown that mineralized dentin particles present intra and extra pores up to 44.48%. This porosity increases the blood supply and supports slow resorption of the grafted material, thereby supporting healing and replacement resorption to achieve lamellar bone [27]. Interestingly, in the present study, socket grafting with MPDA/PRF resulted in a slight increase in both the buccal and lingual bone heights. The most coronal aspect was not on the same level on the buccal or lingual planes at the baseline since the majority of the teeth were extracted due to advanced periodontitis. The most coronal aspects on the buccal and lingual side of the socket at follow-up were determined as coronal reference points regardless of whether it was a cortical bone plate or grafting material still in osseointegration. Figure 4 shows a representative CBCT scan before and after extraction and MPDA/PRF grafting for the periodontally compromised teeth.

Two studies reported a gain in the vertical dimension [20,28]. In a radiographic evaluation after four different grafting materials, +1.2 mm (8.1%) buccal bone height gain after socket grafting with demineralized bovine bone mineral with 10% collagen covered with an autogenous soft-tissue graft was reported [15]. In the other study, dimensional changes were measured on cast models showing the vertical gain both for the non-grafted and grafted sockets [28].

The BPR at 1 mm below the bone crest (BPR-1) was 0.87 mm (± 0.84 mm) in the present study. This amount of resorption corresponds to the measured average initial BBT at this level (0.88 ± 0.49) (BBT-1) and is in accordance with the first bundle bone being resorbed after tooth extraction [4]. The bundle bone is a tooth-dependent structure, and, until now, no grafting material was able to prevent bundle bone resorption.

A ridge width reduction of −1.68 mm ± 1.11 mm for sockets grafted with BioOss Collagen, −1.54 mm for sockets grafted with recombinant human bone morphogenetic protein-2 and demineralized dentin particulate, and −0.78 ± 0.41 mm for the sockets grafted with demineralized dentin particulate measured 1 mm below the buccal bone crest was reported [26]. In the present study, the mean ridge width reduction 1 mm toward the apex from the ridge crest was 1.38 mm ± 1.24 mm. It is important to note that the effectiveness of MPDA in preserving the alveolar ridge was achieved without flap elevation for the primary closure; this simplifies the clinical procedure. Thus, PRF membranes or collagen sponges can be utilized to cover and protect the graft. Primary wound closure may lead to muco-gingival junction repositioning, keratinized mucosa displacement, and additional ridge resorption due to flap elevation [29,30]. All sockets were grafted with a mixture of particulate dentin autograft (2/3) and chopped PRF membranes (1/3). PRF was added to enhance soft tissue healing, facilitate graft handling, and cover the grafted site. However, based on a recent systematic review, the effect of PRF on bone regeneration remains inconclusive [31]. PRF plays a role in accelerating site healing by presenting concentrated amounts of blood derivatives, thereby improving macrophage and growth factor activity, whereas the dentin simultaneously provides the scaffold and long-term maintenance of the site.

The limitations of the study are the retrospective study design, the absence of a control group, and the inclusion of both single and multiple sites. Future research should confirm these initial findings via prospective, randomized, and controlled studies.

## 5. Conclusions

Socket preservation using MPDA with PRF post flapless tooth extraction resulted in well-maintained vertical socket dimensions and minimal horizontal ridge reduction.

## Figures and Tables

**Figure 1 materials-13-01083-f001:**
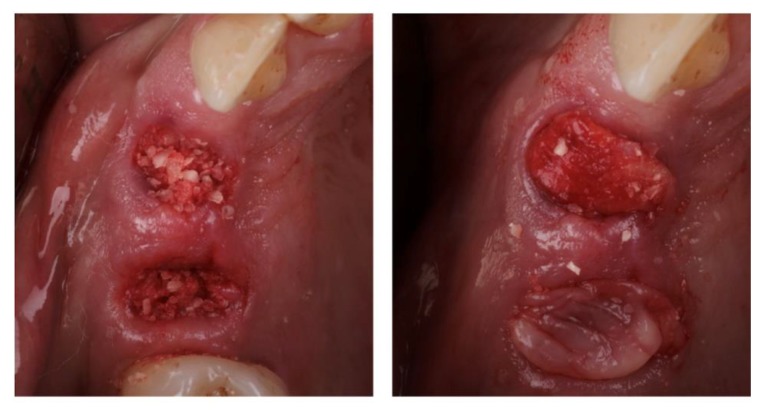
(**a**) A mineralized particulate dentin autograft is mixed with chopped platelet-rich fibrin (PRF) membrane, grafted into the socket; (**b**) graft covered with PRF membrane.

**Figure 2 materials-13-01083-f002:**
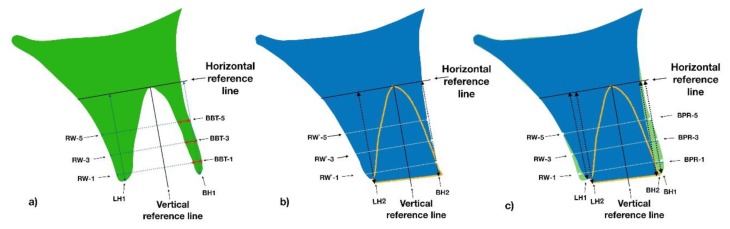
The ridge width was measured 1, 3 and 5 mm below the buccal crest on the initial (**a**) and follow-up cone-beam computed tomography (CBCT) scans (**b**) based on horizontal and vertical reference lines. The ridge height was measured from the most coronal aspects on the buccal and lingual side on the initial (a) and follow-up CBCT scans (**b**). Buccal bone plate reduction was measured on superimposed CBCT scans at three levels below the buccal crest (**c**). RW-1, RW-3, RW-5: initial ridge width at 1, 3, and 5 mm below the buccal crest, respectively; RW`-1, RW`-3, RW`-5: ridge width at 1, 3, and 5 mm, respectively, at follow-up. LH1: initial lingual bone height; LH2: lingual bone height after four months; BH1: initial buccal bone height; BH2: buccal bone height at follow-up; BPR-1, BPR-3, and BPR-5: buccal bone plate reduction at 1, 3, and 5 mm below the buccal crest, respectively.

**Figure 3 materials-13-01083-f003:**
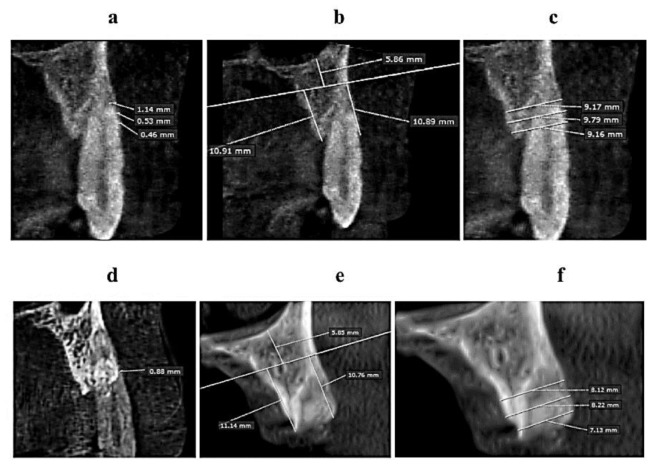
A representative CBCT scan is shown. (**a**) The initial buccal bone plate thickness 1, 3, and 5 mm from the highest buccal ridge point. (**b**) The baseline socket height at the mid-buccal and mid-lingual are shown along with the (**c**) horizontal ridge width 1, 3, and 5 mm from the highest buccal ridge point on the baseline. (**d**) Buccal plate reduction is depicted at three levels measured on superimposed CBCT scans. (**e**) Mid-buccal and mid-lingual bone height on follow-up CBCT scans. (**f**) The horizontal ridge width 1, 3, and 5 mm below the buccal crest on follow-up CBCT scans.

**Figure 4 materials-13-01083-f004:**
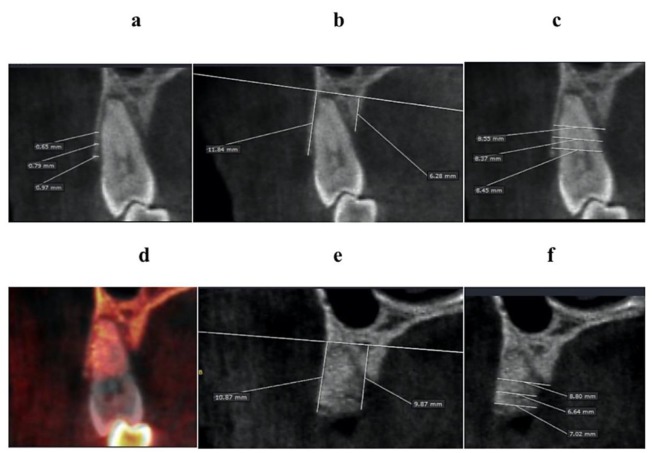
A representative CBCT scan for the case with an increase in lingual bone height, the absence of the buccal bone resorption, and a reduction in ridge width. The first premolar site before tooth extraction and dentin grafting (**a**,**b**,**c**) and after four months is shown (**d**,**e**,**f**). Initially, the lingual bone was 5.58 mm lower than the buccal bone. The lingual bone gained 3.59 mm in height. There is ridge width loss (−1.43 mm, −1.73 mm, and −0.25 mm) at 1, 3, and 5 mm levels. Superimposed CBCT images show that there is no buccal plate reduction.

**Figure 5 materials-13-01083-f005:**
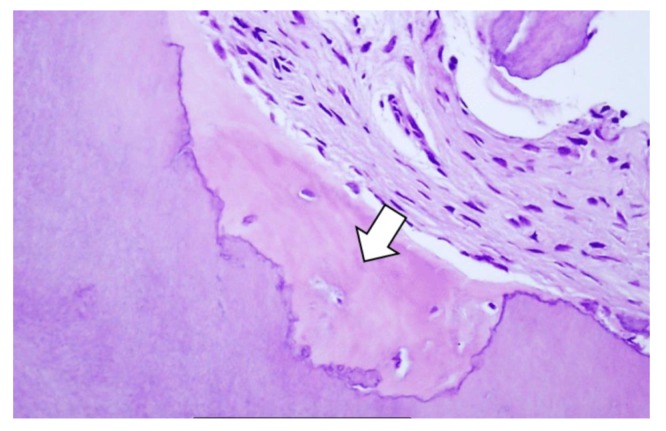
Representative hematoxylin–eosin-stained sample. New bone is formed in direct contact with the dentin particle. Lacunae are filled with cells indicative of live new bone.

**Figure 6 materials-13-01083-f006:**
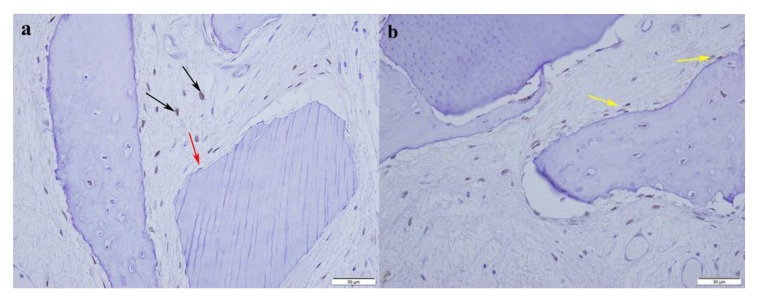
In the contact area between dentin particle and new bone, immunohistochemistry shows the Osterix-positive osteoblasts and pre-osteoblasts in connective tissue (black arrows), on the dentin particle surface (red arrow) (**a**), and in the new formed bone (yellow) (**b**). DAB (3,3′-Diaminobenzidine) was used as a chromogen (Dako), and counterstaining was performed with Shandon hematoxylin.

**Table 1 materials-13-01083-t001:** Measurements of initial buccal bone thickness and horizontal reduction four months after tooth extraction.

Measurements of Buccal Bone		
	Mean (mm)	SD (mm)
Initial thickness at 1 mm below the crest (buccal bone plate thickness (BBT) 1)	0.88	0.49
Initial thickness at 3 mm below the crest (BBT2)	1.11	0.75
Initial thickness at 5 mm below the crest (BBT3)	1.38	1.07
Buccal plate horizontal reduction 1 mm from the highest ridge point (BPR-1) after 4 months	−0.87	0.84
Buccal plate horizontal reduction 3 mm from the highest ridge point (BPR-2) after 4 months	−0.60	0.70
Buccal plate horizontal reduction 5 mm from the highest ridge point (BPR-3) after 4 months	−0.41	0.55

**Table 2 materials-13-01083-t002:** Measurements of ridge width at the time of extraction and four months later.

Ridge Width	Initial		After 4 Months		Difference	
	Mean	SD	Mean	SD	Mean	SD
	(mm)	(mm)	(mm)	(mm)	(mm)	(mm)
Ridge width 1 mm from the highest ridge point	8.45	1.33	7.02	1.42	−1.38	1.24
Ridge width 3 mm from the highest ridge point	8.73	1.35	7.86	1.37	−0.82	1.13
Ridge width 5 mm from the highest ridge point	9.01	1.49	8.55	1.47	−0.43	0.89

**Table 3 materials-13-01083-t003:** Measurements of buccal and lingual bone height at the time of extraction and four months later.

Ridge Height.	Initial		After 4 Months		Difference	
	Mean	SD	Mean	SD	Mean	SD
	(mm)	(mm)	(mm)	(mm)	(mm)	(mm)
Buccal bone height (BH1, BH2, ΔBH)	8.79	2.27	8.96	2.76	+0.16	2.34
Lingual bone height (LH1, LH2, ΔLH)	8.54	2.22	8.79	2.73	+0.4	1.68

**Table 4 materials-13-01083-t004:** Summary of statistical analysis findings for ridge width and height reduction compared to literature reference data.

Measurement ^†^	Hypothetical Mean Basis	Hypothetical Mean ‡	Sample Mean	Sample SD	*t*-test	Significance
Ridge width reduction	Tan et al. [2]; 30%	−2.53	−1.38	1.24	7.063	*p* < 0.0001
	Van der Weijden et al. [3]	−3.87	−1.38	1.24	15.293	*p* < 0.0001
Height loss	Tan et al. [2]; 16%	−1.40	0.16	2.34	4.0357	*p* < 0.0001
	Van der Weijden et al. [3]	−1.67	0.16	2.34	4.9145	*p* < 0.0001

† All measurements except t-test values are in mm. ‡ The hypothetical mean basis was determined based on two of the reference studies highlighted in Section 1. The hypothetical mean basis was determined as the average of the figures these studies provided. In this case, a *t*-test higher than 2.5 suggests strong significance, which means that the difference between the sample and the expected value is significantly different; therefore, the null hypothesis should be rejected.

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
