# Peer review of "Maintenance of Alveolar Ridge Dimensions Utilizing an Extracted Tooth Dentin Particulate Autograft and Platelet-Rich fibrin: A Retrospective Radiographic Cone-Beam Computed Tomography Study"

_materials, 2020, doi:10.3390/ma13051083_

Round 1
Reviewer 1 Report
General comments:
In this paper, the authors reported the effects of particulate dentin autograft mixed with chopped platelet-rich fibrin (PRF) on the changes of alveolar ridge dimensions. Their results showed that the mineralized dentin autograft with PRF is effective in preserving post-extraction alveolar ridge dimensions. This is an interesting study. However, as the authors point out in the discussion section, this study needs a negative control group. And they also need to increase the number of patients.
Author Response
Thank you for your review.
Point one:
This study needs a negative control group.
Answer:
A suitable control group would have been an untreated socket. There are many studies showing ridge volume changes after tooth extraction where socket preservation was not performed. All of these studies are unanimous that untreated sockets will result in reduced alveolar bone dimensions. Therefore we always perform socket preservation procedures as a standard of care. This being a retrospective study, we did not have control data from our patients since all extraction sockets have been treated one way or another. We, therefore, rely on ample data from the literature that describes ridge dimensional changes for untreated sockets.
Point 2:
And they also need to increase the number of patients.
Answer:
We consider 12 patients with 58 sites at different teeth positions, mostly in the esthetic zone, a sufficient number.
Reviewer 2 Report
The current study is on a topic of relevance and general interest to the readers of the journal. It is well organized and the methods were implemented properly. The results are interesting, presented accurately and contribute to the knowledge of the topic of the study. Some suggestions should be the following: In Materials and Methods please provide the number of the extracted incisors, canines and premolars (1st or 2nd), separately. Additionally, please specify which teeth presented periradicular defects before extraction.
Author Response
Thank you for your review and helpful comments.
Point one:
In Materials and Methods please provide the number of the extracted incisors, canines and premolars (1st or 2nd), separately.
Answer:
It is added. Reviewer Nr 4 suggested to move this section from M&M to the Results:
"In the mandible, 19 teeth were extracted, and the sockets were grafted, whereas 39 teeth were extracted in the maxilla. Among the extracted teeth, there were 22 incisors, 12 canines, 9 first premolars, 10 second premolars, and 5 molars."
Point two:
Additionally, please specify which teeth presented periradicular defects before extraction.
It is added. Reviewer Nr 4 suggested to move this section from M&M to the Results:
"Four incisors, 2 premolars, and 1 molar tooth presented periradicular defects."
Reviewer 3 Report
Dear Authors,
congratulations for Your study. I have some minor comments to be addressed to improved the quality of the manuscript.
Introduction: please update literature with more recent in vitro study on the product of smart grinder
https://www.ncbi.nlm.nih.gov/pubmed/30729763
in the discussion the part treating the PRF should be amplified indicating the role of the growth factors which are in the clot (see: http://www.intjmorphol.com/abstract/?art_id=4112)
another point to be added is that is missing the histology of the new formed bone assessing the quality
Author Response
Dear reviewer,
thank you for your encouragement and for your helpful comments.
Point one:
Introduction: please update literature with more recent in vitro study on the product of smart grinder
https://www.ncbi.nlm.nih.gov/pubmed/30729763
Answer:
We added in introduction:
"The human periodontal ligament fibroblast cell line showed a very promising growth reaction to the mineralized dentin in a recently published in vitro study, confirming it`s excellent biocompatibility." (REF)
Point two:
in the discussion the part treating the PRF should be amplified indicating the role of the growth factors which are in the clot (see: http://www.intjmorphol.com/abstract/?art_id=4112)
Answer:
We added:
"As a supplement of bone graft material the use of platelet concentrates such as Platelet Rich Plasma (PRP), Platelet-Rich-Fibrin (PRF), Concentrated Growth Factors (CGF), which regulate inflammation and angiogenesis could be beneficial (REF). These biological additives have been identified as producers of growth factors that promote the wound healing and regenerative process.
In our study a second-generation platelet concentrate - the leucocyte and platelet-rich fibrin (L-PRF) as introduced by Choukroun and co-workers was utilized as supplementary to the particulate dentin graft (REF)"
And:
PRF plays a role in accelerating site healing by presenting concentrated amounts of blood derivatives hence improving macrophages and growth factors activity, whereas, at the same time, the dentin provides the scaffold and long-term maintenance of the site.
Point three:
another point to be added is that is missing the histology of the new formed bone assessing the quality
Answer:
We are planning to write a separate manuscript showing histology, immunohistochemistry, and histomorphometry after dentin grafting. But we completely agree with your suggestion to show the new formed bone quality and included the following:
in "M&M"
Histologic examination
"Histological samples were harvested with a trephine bur during implant preparation, preserved in 10% neutral buffered formalin and sent for histological and immunohistochemical analysis to the Department of Histology and Embryology Faculty of Medicine, University of Rijeka."
In "Results"
"Histological analysis revealed a new bone formation in close contact with dentin particles (ankylosis) with no signs of inflammation or fibrous encapsulation of the autologous augmentation material (Figure 5). New bone was confirmed by osteoblasts marked by antibodies against Osterix (Anti-Sp7/Osterix antibody – Chip Grade, ab22552, Abcam) (Figure 6)."
Figure 5. New bone is formed in direct contact to the dentin particle. Lacunae are filled with cells indicative of live new bone.
Figure 6. In the contact area between dentin particle and new bone, immunohistochemistry is showing the Osterix positive osteoblasts and pre-osteoblasts in connective tissue (black arrows), on the dentin particle surface (red arrow) and in the new formed bone (yellow).
Best regards
Authors

Reviewer 4 Report
I suggest the title be changed to better reflect the work performed. The use of plated-rich fibrin should be included. For example: “Maintenance of Alveolar Ridge Dimensions Utilizing an Extracted Tooth Dentin Particulate Autograft and Plated-rich fibrin: A Retrospective Radiographic Cone-Beam Computed Tomography Study”
The manuscript abstract needs to be rewritten and the English language improved. It lacks the study aim, and the information must be sequentially organized, from material and methods first, to results, and finally discussion and conclusions
Lines 62-64 refers to materials and methods description and should be removed.
Line 70: the authors stated that: “conducted according to the STROBE checklist.” STROBE guidelines provide orientation to trials report as stated at lines 74-75, so this information should be removed here
Please insert the code of the Ethical approval
Lines 81-84 should be moved to the results section since they provide results of the study participants
Lines 91-94 should be moved to the results section since they provide results of the included participants and sites
Several studies report that the use of PRF alone presents benefits in post-extraction sockets preservation. This is a major issue regarding this study. Control groups with PRF alone and dentin autograft alone should be included. Also, the results obtained cannot be attributed only to the dentin autograft. This way, the authors need to include control groups, or to reformulate the manuscript, referring to the effects of dentin autografts and PRF together, as responsible for the results obtained.
Lines 126-127: was the radiographic measurements performed by an external examiner?
Line 131: please refer to figure 3
Lines 140, 143 and 144: figure 3 should not be referred to as CBCT images. It must be referred to as an artificial image or reconstruction based on CBCT images.
Table 1 and Table 2 should be moved to the results section
Tables 1 and 2 would benefit from being rearranged to improve readers' comprehension. They should be simplified, for example presenting the time points in columns and the change at the end of the line.
Lines 191-193: what did the authors mean with “highly significant”?
Please put “et al.” in italic form
Table 3 needs to be improved and moved to the results section. What does sample mean means? It is referent to this study results? The p-value should be added to the table, in the significance column.
Lines 207-220 presented repeated information from the tables. Please choose to present the results in only one form, table or text.
The discussion section needs to take into account that in this study dentin grafts were used together with PRF.
In the conclusion section, the use of PRF must be added to the statement.
Lines 293-295 should be removed or moved to the discussion section since they do not reflect a conclusion of the present work.
Author Response
Dear reviewer,
thank you for your review and helpful comments. We have respected all your advice and changed the manuscript accordingly.
Point one:
I suggest the title be changed to better reflect the work performed. The use of plated-rich fibrin should be included. For example: “Maintenance of Alveolar Ridge Dimensions Utilizing an Extracted Tooth Dentin Particulate Autograft and Plated-rich fibrin: A Retrospective Radiographic Cone-Beam Computed Tomography Study”
Answer:
It is changed. The title: “Maintenance of Alveolar Ridge Dimensions Utilizing an Extracted Tooth Dentin Particulate Autograft and Plated-rich fibrin: A Retrospective Radiographic Cone-Beam Computed Tomography Study”
Point two:
The manuscript abstract needs to be rewritten and the English language improved. It lacks the study aim, and the information must be sequentially organized, from material and methods first, to results, and finally discussion and conclusions
Answer:
The study aim is added:
Abstract: This study utilized radiographic comparative analysis in order to evaluate dimensional ridge changes 4 months after tooth extraction and immediate grafting with mineralized dentin autograft and chopped platelet-rich fibrin. Fifty-eight extraction sockets with up to 2 mm of missing buccal bone in the coronal aspect compared to the lingual bone were included. Graft material was covered with either a platelet-rich fibrin membrane or collagen sponge with no effort to achieve primary closure. The dimensional changes of the ridge were assessed on cone-beam computed tomography images acquired prior to extraction and 4 months later. The reduction in the buccal bone plate thickness 1 mm, 3 mm, and 5 mm below the buccal crest was -0.87 ± 0.84 mm, -0.60 ± 0.70 mm, and -0.41 ± 0.55 mm, respectively. The mean ridge width changes 1 mm, 3 mm, and 5 mm below the crest were -1.38 ± 1.24 mm, -0.82 ± 1.13 mm, and -0.43 ± 0.89 mm, respectively. The average mid-buccal bone height gain was +1.1%, while the mid-lingual height gain was 5.6%. A mineralized dentin autograft with platelet-rich fibrin is effective in preserving post-extraction alveolar ridge dimensions.
Regarding the English language: none of the authors is a native English speaker, but this manuscript is professionally edited, Wiley certificate for advanced language editing is attached.
Point three:
Lines 62-64 refers to materials and methods description and should be removed.
Answer:
This is done.
Point four:
Line 70: the authors stated that: “conducted according to the STROBE checklist.” STROBE guidelines provide orientation to trials report as stated at lines 74-75, so this information should be removed here.
Answer:
This is done.
Point five:
Please insert the code of the Ethical approval
Answer:
Ethical approval for data collection from the Rident clinic, where this retrospective study was conducted, is attached.
Point six:
Lines 81-84 should be moved to the results section since they provide results of the study participants
Answer:
This is done.
Lines 91-94 should be moved to the results section since they provide results of the included participants and sites
Answer:
This is done.
Point seven:
Several studies report that the use of PRF alone presents benefits in post-extraction sockets preservation. This is a major issue regarding this study. Control groups with PRF alone and dentin autograft alone should be included. Also, the results obtained cannot be attributed only to the dentin autograft. This way, the authors need to include control groups, or to reformulate the manuscript, referring to the effects of dentin autografts and PRF together, as responsible for the results obtained.
Answer:
We reformulated the manuscript accordingly and referred to the effects of dentin autografts and PRF together.
Point eight:
Lines 126-127: was the radiographic measurements performed by an external examiner?
Answer:
We included this information:
"All radiographic measurements and analyses were performed by 1 external examiner (WI, 3DVision, Cairo, Egypt)."
Point nine:
Line 131: please refer to figure 3
Answer:
This is done.
"The vertical reference line was outlined from the apex through the center of the socket (Figure 3a)."
Point ten:
Lines 140, 143 and 144: figure 3 should not be referred to as CBCT images. It must be referred to as an artificial image or reconstruction based on CBCT images.
Answer:
This is done.
The ridge width was measured parallel to the horizontal reference line 1 mm, 3 mm, and 5 mm from the highest buccal ridge point on baseline (RW-1, RW-2, RW-3, respectively) and follow-up (RW`-1, RW`-3, and RW`-5, respectively) CBCT images as shown in the drawing (Figure 3a and 3b).
Point eleven:
Table 1 and Table 2 should be moved to the results section
Answer:
This is done.
Point twelve:
Tables 1 and 2 would benefit from being rearranged to improve readers' comprehension. They should be simplified, for example presenting the time points in columns and the change at the end of the line.
Answer:
We changed the tables. Instead of two, there are three tables now and we believe that the changes will improve the reader`s comprehension significantly. Thank you.
Table 1. Measurements of initial buccal bone thickness and horizontal reduction 4 months after tooth extraction.
|
Measurements of Buccal Bone |
|
|
|
|
Mean (mm) |
St. Dev.(mm) † |
|
Initial thickness at 1 mm below the crest (BBT1) |
0.88 |
0.49 |
|
Initial thickness at 3 mm below the crest (BBT2) |
1.11 |
0.75 |
|
Initial thickness at 5 mm below the crest (BBT3) |
1.38 |
1.07 |
|
Buccal plate horizontal reduction 1 mm from the highest ridge point (BPR-1) after 4 months |
- 0.87 |
0.84 |
|
Buccal plate horizontal reduction 3 mm from the highest ridge point (BPR-2) after 4 months |
- 0.60 |
0.70 |
|
Buccal plate horizontal reduction 5 mm from the highest ridge point (BPR-3) after 4 months |
- 0.41 |
0.55 |
† St. Dev, standard deviation
Table 2. Measurements of ridge width at the time of extraction and 4 months later.
|
Ridge Width |
Initial |
After 4 mo. |
Difference |
|||
|
Mean |
St. Dev. |
Mean |
St. Dev. |
Mean |
St. Dev. |
|
|
(mm) |
(mm) |
(mm) |
(mm) |
(mm) |
(mm) |
|
|
Ridge width 1 mm from the highest ridge point |
8.45 |
1.33 |
7.02 |
1.42 |
-1.38 |
1.24 |
|
Ridge width 3 mm from the highest ridge point |
8.73 |
1.35 |
7.86 |
1.37 |
-0.82 |
1.13 |
|
Ridge width 5 mm from the highest ridge point |
9.01 |
1.49 |
8.55 |
1.47 |
-0.43 |
0.89 |
† St. Dev, standard deviation
Table 3. Measurements of buccal and lingual bone height at the time of extraction and 4 months later.
|
Ridge Height |
Initial |
After 4 mo. |
Difference |
|||
|
Mean |
St. Dev. |
Mean |
St. Dev. |
Mean |
St. Dev. |
|
|
(mm) |
(mm) |
(mm) |
(mm) |
(mm) |
(mm) |
|
|
Buccal bone height (BH1, BH2, ΔBH) |
8.79 |
2.27 |
8.96 |
2.76 |
+0.16 |
2.34 |
|
Lingual bone height (LH1, LH2, ΔLH) |
8.54 |
2.22 |
8.79 |
2.73 |
+0.4 |
1.68 |
† St. Dev, standard deviation
Point thirteen:
Lines 191-193: what did the authors mean with “highly significant”?
Answer:
We added:
In this case, A t-test higher than 2.5 suggests strong significance which means that the difference between the sample and the expected value is significantly different, therefore reject the null hypothesis.
Point fourteen:
Please put “et al.” in italic form
Answer:
This is done, both in the table and in text:
|
Measurement † |
Hypothetical mean basis |
Hypothetical mean ‡ |
Sample mean |
Sample St. Dev. § |
t-test ¶ |
Significance |
|
Ridge width reduction |
Tan et al., [2]; 30% |
-2.53 |
-1.38 |
1.24 |
7.063 |
p-Value < 0.0001 |
|
|
Van der Weijden et al., [3] |
-3.87 |
-1.38 |
1.24 |
15.293 |
p-Value<0.0001 |
|
Height loss |
Tan et al., [2]; 16% |
-1.40 |
0.16 |
2.34 |
4.0357 |
p-Value<0.0001 |
|
|
Van der Weijden et al., [3] |
-1.67 |
0.16 |
2.34 |
4.9145 |
p-Value<0.0001 |
Mazor et al. observed the same dynamic
Point fifteen:
Table 3 needs to be improved and moved to the results section. What does sample mean means? It is referent to this study results? The p-value should be added to the table, in the significance column.
Answer:
Table 3 (now 4) is moved. The sample mean is the mean value (average) of the study result.
The p-value is added to the table.
Point sixteen:
Lines 207-220 presented repeated information from the tables. Please choose to present the results in only one form, table or text.
Answer:
We have chosen to present information in the tables, thank you.
Point seventeen:
The discussion section needs to take into account that in this study dentin grafts were used together with PRF.
Answer:
We changed MDPA in MDPA/PRF through the manuscript.
We added in Introduction:
PRF plays a role in accelerating site healing by presenting concentrated amounts of blood derivatives hence improving macrophages and growth factors activity, whereas at the same time the dentin provides the scaffold and long-term maintenance of the site.
We added in Discussion:
PRF plays a role in accelerating site healing by presenting concentrated amounts of blood derivatives hence improving macrophages and growth factors activity, whereas at the same time the dentin provides the scaffold and long-term maintenance of the site.
Point eighteen:
In the conclusion section, the use of PRF must be added to the statement.
Answer:
This is done:
Socket preservation using MPDA with PRF post flapless tooth extraction resulted in well-maintained vertical socket dimensions and minimal horizontal ridge reduction.
Round 2
Reviewer 1 Report
The authors have performed some experiments and revised the manuscript. I think it can be accepted as it is.
Author Response
Thank you for your review.
Best regards
Reviewer 4 Report
Revision manuscript "Maintenance of Alveolar Ridge Dimensions Utilizing An Extracted Tooth Dentin Particulate Autograft: A Radiographic Cone-Beam Computed Tomography Study”
I suggest the title be changed to better reflect the work performed. For example: “Maintenance of Alveolar Ridge Dimensions Utilizing an Extracted Tooth Dentin Particulate Autograft and Plated-rich fibrin: A Retrospective Radiographic Cone-Beam Computed Tomography Study”
Line 16: the word “chopped” is misplaced. I believe the authors want to refer to chopped mineralized dentin autograft and platelet-rich fibrin, instead of “mineralized dentin autograft and chopped platelet-rich fibrin”
Line 53: I suggest that the expression “the human periodontal ligament cell line”, to be replaced by “a human periodontal ligament cell line” or “human periodontal ligament cells”
Line 55: please put “in vitro” in italic form
Line 62: please add a reference at the end of the sentence
Line 66-70: this information should be moved to the discussion section and compared with the obtained results in this study. In the introduction section, it can be referred that previous studies with MPDA were performed, with significant results, however, the % and values of reduction just should be referred at the discussion section, when comparing to this study
Line 71: PRF must be added to study aim, referring to MPDA+PRF
Line 136: figure 3 is referenced before figure 2. The authors must then change the figures 2 and 3 order
Figure 2 caption: reference to BBT, BBT', RW, and RW, should not be used in this figure because the image does not have these abbreviations. The figure caption must be changed, referring just to a representative scan, or the abbreviations need to be added to the figure
Line 185: please remove this line, since the measurements belong to the results section
Line 190: please put “p” in italic form
Lines 195-198: I suggest the histological examination description be moved before the statistical analysis
Lines 195-198: how was histological examination performed? What kind of staining was used? This needs to be described.
Table 4: please remove the word "value" from the p-value. p is sufficient. Please put “p” in italic form
Figure 5 and 6 captions: information about the staining and amplification of the images needs to be added
Author Response
Dear reviewer,
Point one:
“Maintenance of Alveolar Ridge Dimensions Utilizing an Extracted Tooth Dentin Particulate Autograft and Plated-rich fibrin: A Retrospective Radiographic Cone-Beam Computed Tomography Study”
Answer:
This is done. Thank you.
Point 2:
Line 16: the word “chopped” is misplaced. I believe the authors want to refer to chopped mineralized dentin autograft and platelet-rich fibrin, instead of “mineralized dentin autograft and chopped platelet-rich fibrin”
Answer:
Actually we wanted to refer to PRF, but we improved the sentence, by adding "particulate":
"mineralized dentin particulate autograft and chopped platelet-rich fibrin."
Point three:
Line 53: I suggest that the expression “the human periodontal ligament cell line”, to be replaced by “a human periodontal ligament cell line” or “human periodontal ligament cells”
Answer:
This is done:
"A human periodontal ligament fibroblast cell line..."
Point four:
Line 55: please put “in vitro” in italic form
Answer:
This is done:
published in vitro study
Point five:
Line 62: please add a reference at the end of the sentence
Answer:
This is done, thank you.
"...promote the wound healing and regenerative process [15, 16].
Point six:
Line 66-70: this information should be moved to the discussion section and compared with the obtained results in this study. In the introduction section, it can be referred that previous studies with MPDA were performed, with significant results, however, the % and values of reduction just should be referred at the discussion section, when comparing to this study
Answer:
This is done.
Introduction:
Recently, a human pilot study compared dimensional ridge changes 8 and 16 weeks after tooth extraction and either filling the socket with a mineralized particulate dentin autograft (MPDA) (test group) or without grafting (control group), showing significantly better results for the test group[18].
!!!!!
Point seven:
Line 71: PRF must be added to study aim, referring to MPDA+PRF
Answer:
This is done:
"The aim of this study was to evaluate MPDA/PRF scaffold properties on a larger number of post-extraction sites than that of previous studies."
Point eight:
Line 136: figure 3 is referenced before figure 2. The authors must then change the figures 2 and 3 order
Answer:
This is done.
Point nine:
Figure 2 caption: reference to BBT, BBT', RW, and RW, should not be used in this figure because the image does not have these abbreviations. The figure caption must be changed, referring just to a representative scan, or the abbreviations need to be added to the figure
Answer:
This is done (Figure 2 became Figure 3)
"Figure 3. A representative CBCT scan is shown. (a) The initial buccal bone plate thickness 1, 3, and 5 mm from the highest buccal ridge point. (b) The baseline socket height at the mid-buccal and mid-lingual are shown along with the (c) horizontal ridge width 1, 3, and 5 mm from the highest buccal ridge point on the baseline. (d) Buccal plate reduction is depicted at 3 levels measured on superimposed CBCT scans. (e) Mid-buccal and mid-lingual bone height on follow-up CBCT scans (d) The horizontal ridge width 1, 3, and 5 mm below the buccal crest on follow-up CBCT scans. CBCT: cone-beam computed tomography."
Point ten:
Line 185: please remove this line, since the measurements belong to the results section
Answer:
This is done.
Point eleven:
Line 190: please put “p” in italic form
Answer:
This is done.
of 95% (p < 0.05)
Point twelve:
Lines 195-198: I suggest the histological examination description be moved before the statistical analysis
Answer:
This is done.
Point thirteen:
Lines 195-198: how was histological examination performed? What kind of staining was used? This needs to be described.
Answer:
This is described:
Surgically obtained parts of alveolar bone were fixed immediately in 4% paraformaldehyde for 24 hours and decalcified in Osteodec (Bio Optica 05-MO3005) for 6 days at room temperature.
Following several rinses in PBS specimens were dehydrated through an alcohol sequence, cleared in xylene and embedded in paraffin. Microtome sections (3μm) were intended for histological and immunohistochemistry analysis.
The morphology of the dental and bone tissues was qualitatively evaluated on the hematoxylin and eosin-stained slides.
Previously to the immunohistochemistry staining, endogenous peroxidase activity was inhibited by Dako Peroxidase – Blocking solution for 5 minutes at room temperature. Subsequently, sections were subjected to antigen retrieval with citrate phosphate buffer (pH 6.0).
To evaluate osteoblast differentiation and bone formation, primary antibodies against Osterix (Anti-Sp7/Osterix antibody – Chip Grade, ab22552, Abcam) were used, followed by polyclonal goat anti-rabbit/biotinylated secondary antibody (E 0432; Dako, Denmark) and Streptavidin POD conjugate (Roche Diagnostic GmbH; Germany). DAB was used as a chromogen (Dako) and counterstaining was performed with Shandon hematoxylin.
Slides were analyzed on an Olympus BX51 microscope and images were acquired by Olympus digital camera DP71.
Point fifteen:
Table 4: please remove the word "value" from the p-value. p is sufficient. Please put “p” in italic form
Answer:
This is done.
Point sixteen:
Figure 5 and 6 captions: information about the staining and amplification of the images needs to be added
Answer:
Caption Figure 5:
"Representative hematoxylin-eosin stained sample. New bone is formed in direct contact with the dentin particle. Lacunae are filled with cells indicative of live new bone"
Caption Figure 6:
Figure 6. In the contact area between dentin particle and new bone, immunohistochemistry is showing the Osterix positive osteoblasts and pre-osteoblasts in connective tissue (black arrows), on the dentin particle surface (red arrow) and in the new formed bone (yellow). DAB was used as a chromogen (Dako) and counterstaining was performed with Shandon hematoxylin.
Regarding amplification: Prof Tomac used the objective magnification 20X, but I am not sure if it is the information that you asked for.
Thank you for your review.